# Effect of Groundwater Level Rise on the Critical Velocity of High-Speed Railway

Jing Hu [1,*], Linlian Jin [1], Shujing Wu [1], Bin Zheng [2], Yue Tang [3] and Xuezheng Wu [1]

[1] College of Civil Engineering, Fuzhou University, Fuzhou 350108, China; 220527102@fzu.edu.cn (L.J.); 052001202@fzu.edu.cn (S.W.); wu@fzu.edu.cn (X.W.)
[2] Huadong Engineering Corporation Limited, Fuzhou 350001, China; zheng_b2@hdec.com
[3] Communications Design Research Institute Co., Ltd. of Jiangxi Province, Nanchang 330002, China; yuetang@proton.me
[*] Correspondence: jingh@fzu.edu.cn

**Abstract:** The increasing frequency of extreme rainfall is leading to a rise in groundwater levels in coastal areas, significantly affecting high-speed railway operations. To address this concern, this study developed a 2.5-dimensional finite element model of a coupled track-embankment-ground system based on Biot's porous media theory to analyze the effect of groundwater level rise on the critical velocity of high-speed railways and vibration responses. The findings reveal a consistent decrease in the critical velocity of high-speed railways with rising groundwater levels. Particularly, the increase in groundwater levels within the embankment significantly influences the critical velocity compared to a similar rise in the foundation's groundwater level. Furthermore, deformations induced by passing trains significantly increase as groundwater levels rise. Specifically, when the groundwater level rises from the foundation bottom to the subgrade surface, subgrade surface deformation increases by approximately 55%. As trains approach the critical velocity, significant vibration phenomena, known as the "Mach effect," occur at the foundation surface. Importantly, as groundwater levels rise, the "Mach effect" intensifies. Analyzing the vibrating frequency spectrum of the displacement response demonstrates a substantial increase in vibration amplitude, particularly in the high-frequency region, as groundwater levels rise. This study highlights that the rise in groundwater level not only amplifies vibrations but also extends the propagation of high-frequency vibrations, underscoring the importance of effective embankment waterproofing in controlling track vibrations.

**Keywords:** high-speed railway; critical velocity; groundwater level rise; 2.5D FEM; dynamic response

## 1. Introduction

In recent years, southern China has frequently experienced extreme rainstorms, leading to a significant increase in groundwater storage in the Yangtze River basin [1,2]. As the water content of embankment and foundation soils increases, and soil strength decreases, the risk of train derailment and other hazards becomes more significant, particularly when the train runs at high speed. This is because the "resonance" phenomenon may occur in the railway embankment–foundation system, resulting in strong vibration, and the speed at which this phenomenon occurs is defined by scholars as the "critical velocity" [3,4]. Therefore, studying the effect of groundwater level rise on the critical velocity of the high-speed railway and the dynamic response under critical velocity is of utmost practical significance.

During a passage of the X2000 high-speed train through the Ledsgard area at a speed of 200 km/h, the maximum vibration displacement of the track exceeded the vibration limit specified for safe operation, reaching up to 14 mm [5]. This incident marked the first identification of the critical velocity problem caused by train operation. Eventually, the issue was resolved through a combination of speed reduction and foundation reinforcement [5]. Researchers conducted studies based on the critical velocity problem of the X2000. Auersch [6] discovered that the dynamic response of the soft foundation significantly

increases when the train runs at critical velocity. Additionally, Bian et al. [7] conducted a 2.5-dimensional finite element method (2.5D FEM) to investigate the foundation vibration problem under high-speed train load. By analyzing different numerical models, Costa et al. [4] found that the critical velocity of the railway is determined by the performance of the track, subgrade, and foundation. Fernández-Ruiz et al. [8] explored the non-linear reduction in soil stiffness during loading and delved into the impact of soil non-linearity on the critical velocity of concrete slabs and ballasted tracks. In efforts to manage long-term settlement, numerous high-speed railways have adopted reinforced scenarios to enhance foundation soil. Alexandre et al. [9,10] utilized three-dimensional periodic modeling to investigate the critical velocity of reinforced soils. In recent years, scholars have considered the fluid-solid coupling effect in actual soil and used two-phase saturated media to simulate the foundation soil [11–15]. Among them, Bian et al. [11] and Hu et al. [12] established a train-subgrade-saturated foundation model, which found that when the train moves at critical velocity, it not only causes strong vibration displacement but also excites significant excess pore pressure.

Natural factors, such as rainfall and fluctuations in groundwater levels, can cause changes in the water content of the foundation and embankment, leading to a variation in their mechanical properties, which in turn affects the critical velocity and dynamic response characteristics of the entire embankment–foundation system when subjected to high-speed train load [16]. Jiang et al. [17] investigated the impact of groundwater level rise on the dynamic performance of the embankment–foundation system based on a full-scale physical model of a ballastless track. Similarly, Chen et al. [18] studied the relationship between groundwater level rise and subgrade deformation based on a full-scale physical model. Chen et al. [19] investigated the dynamic response characteristics of the high-speed train subgrade under rainwater infiltration by conducting large-scale field tests.

However, previous studies have not addressed the effect of groundwater level rise on the critical velocity, as large model tests and field tests cannot be loaded at ultra-high speed. Therefore, in this paper, a 2.5D finite element analysis model of track–embankment–foundation dynamic coupling consistent with the actual embankment dimensions is established based on Biot's saturated porous media theory. The model investigates the effect of groundwater level rise in the foundation and embankment on the critical velocity and analyzes the dynamic response of the system when the train runs at speeds less than, equal to, and greater than the critical velocity with the consideration of different groundwater levels.

## 2. Theoretical Solution Method

### 2.1. Biot's Porous Media Theory

The dynamic control equations for saturated porous media in *u-w* format [20]:

$$\mu u_{i,jj} + (\lambda + \alpha^2 M + \mu)u_{j,ji} + \alpha M w_{j,ji} = \rho_b \ddot{u}_i + \rho_f \ddot{w}_i \tag{1}$$

$$\alpha M u_{j,ji} + M w_{j,ji} = \rho_f \ddot{u}_i + m\ddot{w}_i + b\dot{w}_i \tag{2}$$

where $u_i$ and $w_i$ represent soil skeleton displacement and pore water displacement relative to soil skeleton, respectively; $\rho_b = n\rho_f + (1-n)\rho_s$ is the density of saturated soil, where $\rho_s$ and $\rho_f$ are the density of soil particles and the density of pore fluid, respectively; $n$ is the porosity; $m$ is the effective density, $m = a_\infty \rho_f / n$, where $a_\infty = 1/\sqrt{n}$ is a measure of soil pore curvature; $b = \rho_f g/k_D$, $b$ represents the viscous coupling between pore fluid and soil particles, $k_D$ is the Darcy permeability coefficient of saturated soil in m/s; $g$ is gravitational; $\alpha$, $M$ is the Biot constant, where $\alpha = K/K_s$, $\frac{1}{M} = (n/K_f) + (\alpha - n/K_s)$, $K$, $K_s$, and $K_f$ are the bulk modulus of the soil skeleton, soil particles, and pore fluid, respectively; subscripts $i, j = x, y, z$ are tensor notations; $\lambda$ is the first parameter of Lame of the soil skeleton, $\mu$ is the second parameter of Lame of the soil skeleton; superscripts '·', '··', represent the first- and second-order derivatives with respect to time, respectively.

### 2.2. 2.5D Finite Element Solution

Define the Fourier transform concerning time and space:

$$\widetilde{\overline{h}}(\xi_x, y, z, \omega) = \int_{-\infty}^{+\infty}\int_{-\infty}^{+\infty} h(x, y, z, t)e^{i\xi_x x}e^{-i\omega t}\mathrm{d}x\mathrm{d}t \tag{3}$$

The corresponding Fourier inverse transform is:

$$h(x, y, z, t) = \frac{1}{4\pi^2}\int_{-\infty}^{+\infty}\int_{-\infty}^{+\infty} \widetilde{\overline{h}}(\xi_x, y, z, \omega)e^{i\xi_x x}e^{i\omega t}\mathrm{d}\xi_x\mathrm{d}\omega \tag{4}$$

where $x$ represents the direction of train movement along the track; $t$ represents the time; $\xi_x$ represents the wave number along the $x$ direction; $\omega$ represents the circular corner frequency. $h$ is a variable in the space and time domain and $\widetilde{\overline{h}}$ is a variable in the wave number and frequency domain.

By substituting Equation (3) into Equations (1) and (2), the Biot dynamic control equation in the frequency domain-wave number domain is obtained. Referring to Bian et al. [11], the final matrix equation for the 2.5D finite element solution is obtained:

$$(K_1 + K_2 - M_1)\widetilde{\overline{U}} + (L_1 - M_2)\widetilde{\overline{W}} = \widetilde{\overline{F}}^s \tag{5}$$

$$(K_3 - M_3)\widetilde{\overline{U}} + (L_2 - M_4)\widetilde{\overline{W}} = \widetilde{\overline{F}}^f \tag{6}$$

where $M$ represents the mass matrix; $K$ and $L$ represent the stiffness matrices; $U$ and $W$ represent the soil skeleton displacement matrix and the pore fluid relative displacement matrix, respectively; $F^s$ and $F^f$ represent the external load vectors.

### 2.3. Train-Track-Embankment Coupling

In this process, the embankment and foundation are treated as porous media. The track is simplified as a Euler beam placed on the embankment. According to the Euler beam dynamic equation, the vibration of the track under the action of the wheel–rail force can be described as:

$$\left(K_T - \omega^2 M_T\right)\widetilde{\overline{U}}_T = \widetilde{\overline{F}}_{IT} + \widetilde{\overline{P}}_k \tag{7}$$

where $K_T$ and $M_T$ represent the stiffness matrix and mass matrix of the rail; $\widetilde{\overline{F}}_{IT}$ is the supporting force at the embankment surface; $\widetilde{\overline{P}}_k$ represents the force vector of the train load on the surface of the rail, which is usually calculated by the quarter car model [21]. Figure 1 shows the geometric profile of train wheel loads of k carriages.

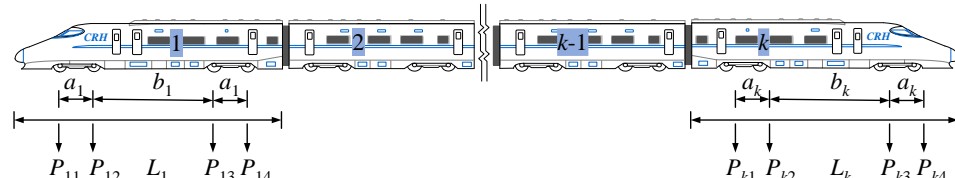

**Figure 1.** Geometric profile of train wheel loads of k carriages.

According to the quarter car model [21], the expression of the force vector of the train load on the surface of the rail in wave number and frequency domain is [22]:

$$\widetilde{\overline{P}}_k(\xi_x, \omega) = \sum_{n=1}^{k}\sum_{i=1}^{4}\left(e^{-i\xi_x x_{ni}}\right)\widetilde{\overline{P}}_{ni}(\xi_x, \omega) \tag{8}$$

where $P_{ni}$ is the axle load for $i$th wheelsets of the $n$th carriage. As indicated in Figure 1, $L_n$ is the length of the $n$th carriage, $a_n$ is the length of the bogie wheelbase, and $b_n$ is the distance from the second to third axles of the carriage of the $n$th carriage. $x_{n1} = \sum_{n=1}^{k} L_n + x_0$; $x_{n2} = a_n + \sum_{n=1}^{k} L_n + x_0$; $x_{n3} = a_n + b_n + \sum_{n=1}^{k} L_n + x_0$; $x_{n4} = 2a_n + b_n + \sum_{n=1}^{k} L_n + x_0$, among which $x_0$ is the distance to the first axle load position.

Based on the deformation compatibility conditions and force equilibrium conditions of the interaction points on the contact surface of the track and the embankment surface, the dynamic equations of the whole vehicle–track–embankment coupling system are obtained as follows [12]:

$$\begin{bmatrix} K_1 + K_2 + K_\mathrm{T} - \omega^2(M_1 + M_\mathrm{T}) & L_1 - \omega^2 M_2 \\ K_3 - \omega^2 M_3 & L_2 - \omega^2 M_4 \end{bmatrix} \begin{bmatrix} \widetilde{\overline{U}} \\ \widetilde{\overline{W}} \end{bmatrix} = \begin{bmatrix} \widetilde{\overline{P}}_\mathrm{M} \\ \widetilde{\overline{F}}^\mathrm{f} \end{bmatrix} \tag{9}$$

The expressions of each matrix in Equation (9) are shown in the literature [12].

### 2.4. Model Validation

Jiang et al. [17] constructed a full-scale model of a slab track–embankment system following the design specifications for Chinese high-speed trains. Figure 2 illustrates a standard high-speed railway line in China in a realistic setting. Figure 3 provides a schematic diagram of the full-scale model and the test cases conducted in Jiang's study. In his test, the groundwater level in Case 1 is at the bottom of the foundation soil; with the water rise, the groundwater level for Case 2 is at the surface of the foundation and subgrade, respectively. In Case 4, the groundwater level returns to the surface of the foundation, enabling an exploration of the effects of the drying and watering cycle on the subgrade. In Jiang's full-scale model test [17], the dynamic performances of this track–embankment system under several typical groundwater levels were measured. In this section, measurements of vibrations at the embankment surface in Cases 2 and 3 were selected to validate the simulation results. As shown in Figure 4, the vibration velocity of the embankment surface calculated by the 2.5D FEM is compared with the model test results. The comparison reveals that, in the range of train speeds from 1 m/s to 100 m/s, the 2.5D finite element calculation results match the test results, verifying the accuracy of the 2.5D finite element model.

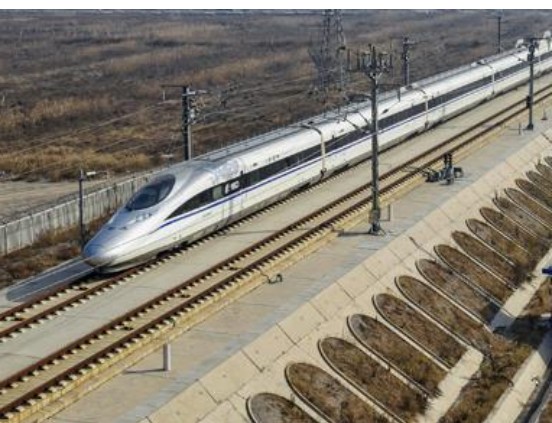

**Figure 2.** A standard high-speed railway line in China.

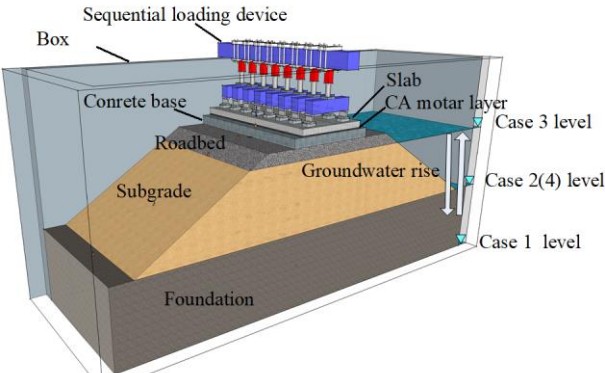

**Figure 3.** Schematic diagram of the full-scale model and test cases in Jiang's research.

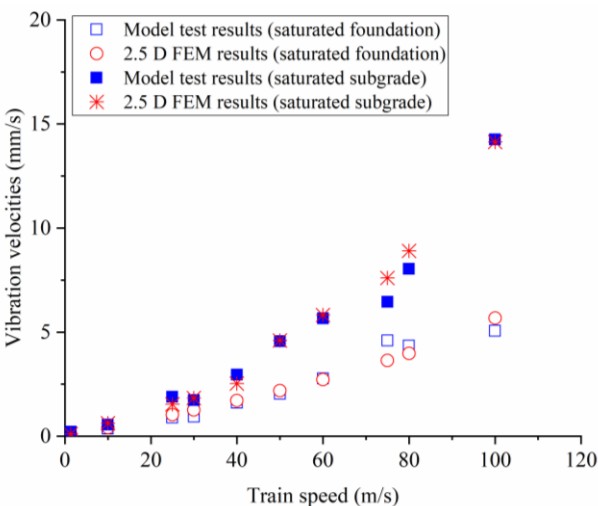

**Figure 4.** Verification of 2.5D element method via model test results.

## 3. Numerical Modelling

### 3.1. Introduction of the Model

In this section, a 2.5D finite element model illustrating the track–embankment–foundation coupling system is depicted in Figure 5. The model possesses a total depth of 63 m and a width of 100 m. The embankment is 3 m thick, encompassing a 0.4 m roadbed and a 2.6 m subgrade. The foundation is structured with an upper permeable layer (0.5 m thick), followed by soil layers: soil layer 1 (3 m thick), soil layer 2 (3 m thick), soil layer 3 (12 m thick), and a lower permeable and damping layer (24.5 m thick), arranged in order from top to bottom. A 2.5D saturated linear elastic element with six degrees of freedom is utilized to simulate the embankment, soil layers, permeable layer, and damping layer. By selecting parameters $\alpha$, $M$, $n$, and $m$ to be close to 0, the saturated two-phase medium can be simplified into a single-phase medium to simulate the soil above the groundwater level. The model incorporates five observation points, designated as A to D. Point A is situated on the surface of the embankment, point B resides on the surface of the foundation, point C is positioned at the foot of the embankment, and point D is located on the ground, 10 m away from the center of the embankment.

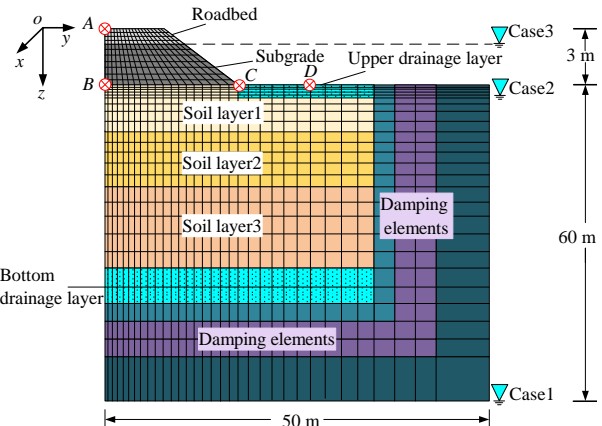

**Figure 5.** Schematic view of numerical model (not scaled).

*3.2. Calculated Cases*

As shown in Figure 5, three different groundwater levels were used in the numerical simulation to replicate the effects of water level rise in the foundation and embankment. The groundwater level in Case 1 was set at the bottom of the foundation, while Case 2 had the groundwater level at the surface of the foundation. Case 3 had the groundwater level located at the surface layer of the subgrade. The initial and saturated values of the soil materials for each layer of the embankment and foundation are provided in Tables 1 and 2, respectively. Table 3 presents the track parameters used in the simulation. The vehicle loads in the numerical simulation were based on the CRH2 train, and a quarter car model was employed for the simulation. The parameters of CRH2 are provided in Table 4.

**Table 1.** Parameters used for each initial soil layer.

| Soil Layer | Biot's Constant $\alpha$ | Biot's Constant $M$ (MPa) | Young's Modulus $E$ (MPa) | Poisson's Ratio $v$ | Density of Soil Particles (kg/m$^3$) | Liquid Density (kg/m$^3$) | Soil Damping $D_0$ | Porosity $n$ | Permeability Coefficient $k_D$ (m/s) |
|---|---|---|---|---|---|---|---|---|---|
| Roadbed | 0.001 | 0.001 | 240 | 0.25 | 2500 | 0.001 | 0.05 | 0.001 | $10^{-20}$ |
| Subgrade | 0.001 | 0.001 | 140 | 0.3 | 2200 | 0.001 | 0.05 | 0.001 | $10^{-20}$ |
| Soil layer 1 | 0.001 | 0.001 | 113 | 0.35 | 2700 | 0.001 | 0.05 | 0.001 | $10^{-20}$ |
| Soil layer 2 | 0.001 | 0.001 | 113 | 0.35 | 2700 | 0.001 | 0.05 | 0.001 | $10^{-20}$ |
| Soil layer 3 | 0.001 | 0.001 | 135 | 0.35 | 2700 | 0.001 | 0.05 | 0.001 | $10^{-20}$ |

**Table 2.** Parameters used for each saturated soil layer.

| Soil Layer | Biot's Constant $\alpha$ | Biot's Constant $M$ (MPa) | Young's Modulus $E$ (MPa) | Poisson's Ratio $v$ | Density of Soil Particles (kg/m$^3$) | Liquid Density (kg/m$^3$) | Soil Damping $D_0$ | Porosity $n$ | Permeability Coefficient $k_D$ (m/s) |
|---|---|---|---|---|---|---|---|---|---|
| Roadbed | 0.001 | 0.001 | 240 | 0.25 | 2500 | 1000 | 0.05 | 0.001 | 1 |
| Subgrade | 1.000 | 6400 | 80 | 0.3 | 2700 | 1000 | 0.05 | 0.3 | $10^{-6}$ |
| Soil layer 1 | 1.000 | 3520 | 45 | 0.35 | 2700 | 1000 | 0.05 | 0.6 | $10^{-6}$ |
| Soil layer 2 | 1.000 | 3520 | 45 | 0.35 | 2700 | 1000 | 0.05 | 0.6 | $10^{-8}$ |
| Soil layer 3 | 1.000 | 3520 | 60 | 0.35 | 2700 | 1000 | 0.05 | 0.6 | $10^{-6}$ |

**Table 3.** Parameters of slab track.

| Rail Mass per Linear Meter (kg/m) | Rail Bending Stiffness (MNm$^2$) | Slab Bending Stiffness (MNm$^2$) | Mass per Linear Meter of Slab (kg/m) | Stiffness of CA Mortar Layer (MN/m/m) |
|---|---|---|---|---|
| 60.64 | 6.625 | 40 | 950 | 100 |

**Table 3.** *Cont.*

| Rail Mass per Linear Meter (kg/m) | Rail Bending Stiffness (MNm²) | Slab Bending Stiffness (MNm²) | Mass per Linear Meter of Slab (kg/m) | Stiffness of CA Mortar Layer (MN/m/m) |
|---|---|---|---|---|
| Damping of CA mortar layer (Ns/m/m) | Bending stiffness of the concrete base (MNm²) | Mass per linear meter of the concrete base (kg/m) | Fastener stiffness (MN/m/m) | Fastener damping (Ns/m/m) |
| $2 \times 10^5$ | 190 | 1800 | 28.5 | $5 \times 10^4$ |

**Table 4.** Parameters used for CRH2 train.

| Parameter Name | Value |
|---|---|
| Carriage mass/kg | 45,000 |
| Bogie mass/kg | 3600 |
| Wheelset quality/kg | 1700 |
| Carriage length/m | 24.8 |
| Centre-to-centre distance of adjacent bogies/m | 14.9 |
| Bogie length/m | 2.5 |

## 4. Numerical Analysis

### 4.1. Critical Velocity

The high-speed railway is composed of the track structure, embankment, and foundation in a top-down arrangement. Figure 6 shows the development curves of the maximum displacement with the train speeds at observation points A and B under different groundwater levels. Point A represents the vibration of the entire track–embankment–foundation system, and its critical velocity is known as the system critical velocity ($V_s$). On the other hand, point B reflects the intensity of the surface vibration of the foundation, and its critical velocity is referred to as the foundation critical velocity ($V_g$).

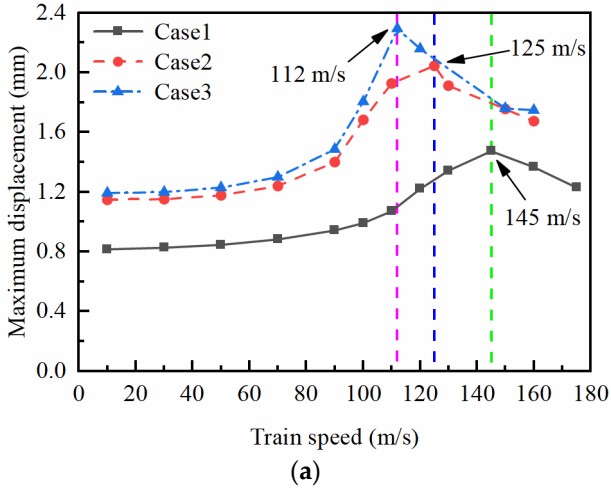

**(a)**

**Figure 6.** *Cont.*

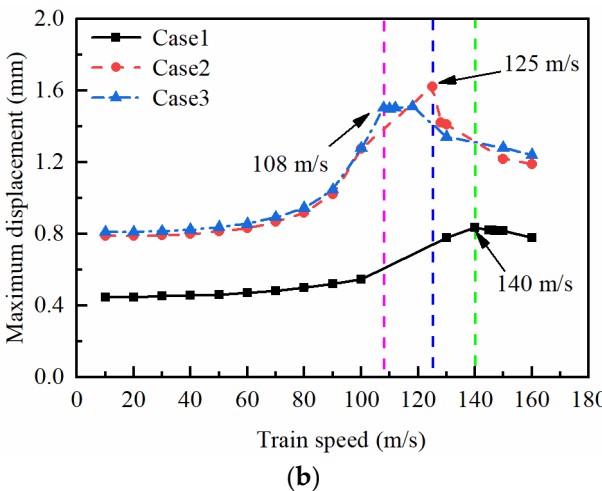

(**b**)

**Figure 6.** Development of the maximum vertical displacements versus train speed at different depths under different cases. (**a**) Point A; (**b**) Point B.

From Figure 6, it is evident that the displacement response amplitudes at different depths in different cases are affected by train speed. The amplitude increases with train speed when the critical velocity has not been reached but decreases when the train speed exceeds the critical velocity. The stiffness of the track structure and embankment is typically greater than that of the foundation, resulting in a system critical velocity ($V_s$) being higher than the critical velocity of the foundation ($V_g$) under the same groundwater levels. By comparing the system critical velocities under different groundwater levels in Figure 6a, it can be seen that $V_s$ decreases as the groundwater level rises. Specifically, for Case 1, $V_s$ is 145 m/s, while for Case 2 (groundwater level rising to the surface of the foundation), $V_s$ decreases by 13.8% to 125 m/s. When the groundwater level is further raised to the surface of the subgrade (Case 3), $V_s$ reduces by 22.8% to 112 m/s as compared to Case 1. These findings indicate that the increase in groundwater level in the embankment has a more significant impact on $V_s$ than that on the foundation. While the foundation critical velocity $V_g$ is slightly less affected by the groundwater level rise, it still decreased by 10.7% and 22.8% when the groundwater level was raised from Case 1 to Case 2 and Case 3, respectively. Even though the vibration intensity of Case 3 is lighter than Case 2, the elevation of the groundwater level in the embankment still further reduces $V_g$.

To provide further insight into the effects of train speed and groundwater level on vibration displacement, Figure 7 displays the vibration displacement time–history curves at point A and point B under each Case. These curves correspond to the system critical velocity ($V_s$) and foundation critical velocity ($V_g$) for each Case, respectively.

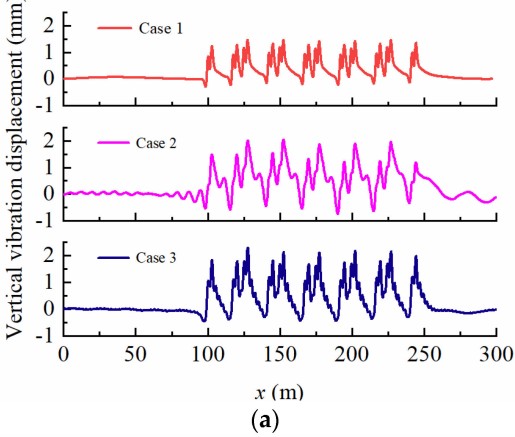

(**a**)

**Figure 7.** *Cont.*

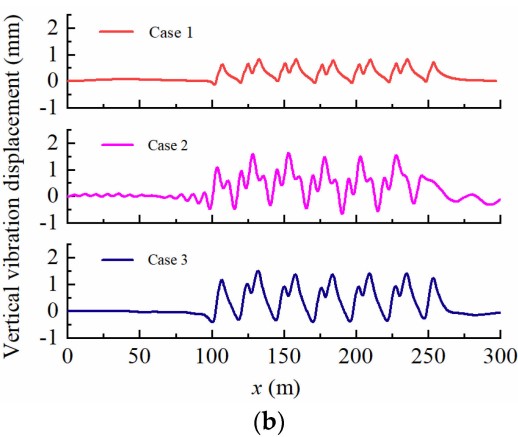

**Figure 7.** Time–history curves at points A and B of different cases. (**a**) Point A; (**b**) Point B.

From Figure 7a, it can be observed that the displacement response amplitudes at point A follow a similar trend across all Cases, but the shape of the displacement response is influenced by the groundwater levels. When the groundwater level is at the bottom of the foundation (Case 1), the displacement response time–history curve displays a "peak column" at the train axle, with a maximum displacement response value of 1.48 mm. When the groundwater level is at the surface of the foundation (Case 2), the shape of the displacement response is similar to that of Case 1 but with a maximum displacement response value of 2.05 mm, representing a 38.5% increase compared to Case 1. When the groundwater level is at the surface of the subgrade (Case 3), the displacement time–history curve at the train axle not only displays a "peak column" but also exhibits fluctuation. The maximum displacement response value in Case 3 is 2.3 mm, representing a 55.4% increase compared to Case 1. These findings indicate that the increase in groundwater level within the embankment causes more violent vibrations. Based on Figure 7b, it is evident that the displacement response amplitude at point B follows a similar development trend as that at point A under the critical velocities of each Case. The groundwater level also has an impact on the foundation displacement response, with the maximum displacement response being 0.833 mm for Case 1 when the groundwater level is at the bottom of the foundation. For Case 2, the maximum displacement response is 1.62 mm, which represents a 94.5% increase compared to Case 1. Similarly, for Case 3, the maximum displacement response is 1.50 mm, which is an 80.1% increase compared to Case 1. Comparing Figure 7a,b, it can be inferred that the vibration displacement response at the surface of the embankment is more pronounced than that at the surface of the foundation when the same groundwater level is present.

*4.2. Dynamic Response in the Foundation*

Figure 8 shows the three-dimensional distribution of the displacement responses at the foundation surface when the train is running at 100 m/s under different groundwater levels. Figure 9 shows the corresponding displacement responses at the *x–y* plane.

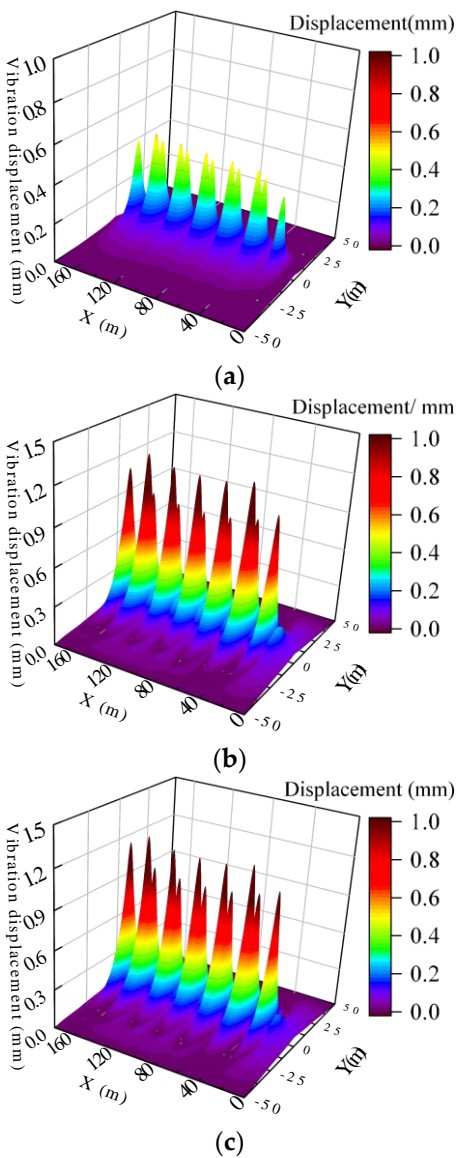

**Figure 8.** Three-dimensional displacement contours at the foundation surface. (**a**) Case 1; (**b**) Case 2; (**c**) Case 3.

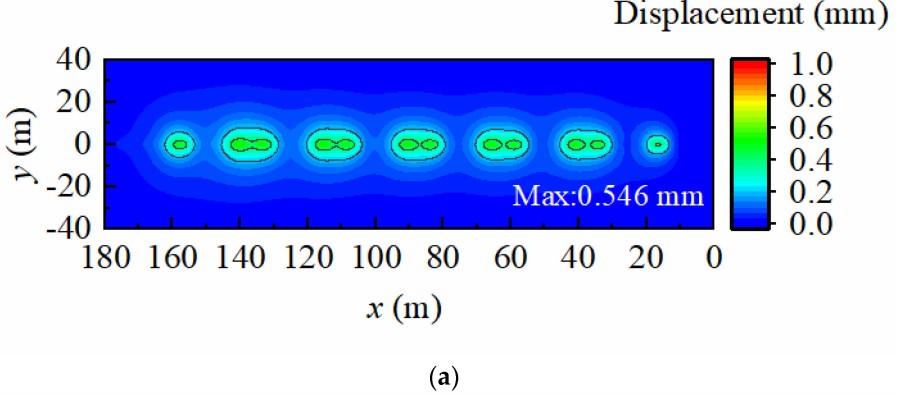

(**a**)

**Figure 9.** *Cont*.

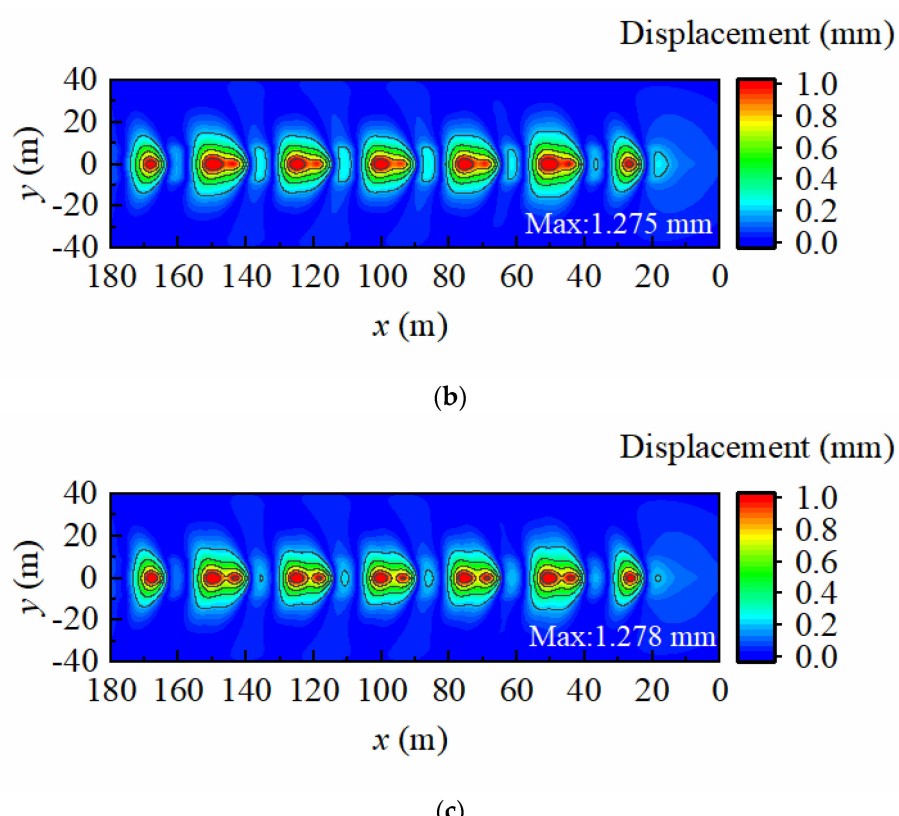

**Figure 9.** Displacement contours in $x$–$y$ plane. (**a**) Case 1; (**b**) Case 2; (**c**) Case 3.

Based on Figure 8, it is clear that the foundation surface undergoes significant deformation under different groundwater levels when the train is running at 100 m/s. Additionally, small-scale regional vibrations occur in the area in front of and behind the axle action position. In Case 1, the displacement vibration on the foundation surface is distributed correspondingly with the train load. In Case 2, the displacement vibration on the foundation surface is not limited to the range corresponding to the train load but shows an obvious wave propagation phenomenon. Moreover, very noticeable displacement vibrations occur around the train load. In Case 3, the fluctuation around the train load reduces. From Figure 9, the surface deformation is observed as a fluctuating line extending rearward from the position of the train load action. Narrow wings are also evident in the distribution of the foundation displacement response behind the train load, resulting in a Mach cone and the apparent Mach effect phenomenon. The critical velocity of the system in Case 2 is 125 m/s, which is the closest to that of the foundation. Thus, the surface vibration of the foundation in Case 2 is the most prominent in Figures 8 and 9.

*4.3. Displacement Response Spectrum*

Figure 10 shows the frequency spectra curves of the displacement response at observation points A, C, and D under Case 3, with consideration to three train operating speed conditions: 0.5 $V_s$, 1 $V_s$, and 1.2 $V_s$. In this context, the value of $V_s$ in Case 3 is 112 m/s.

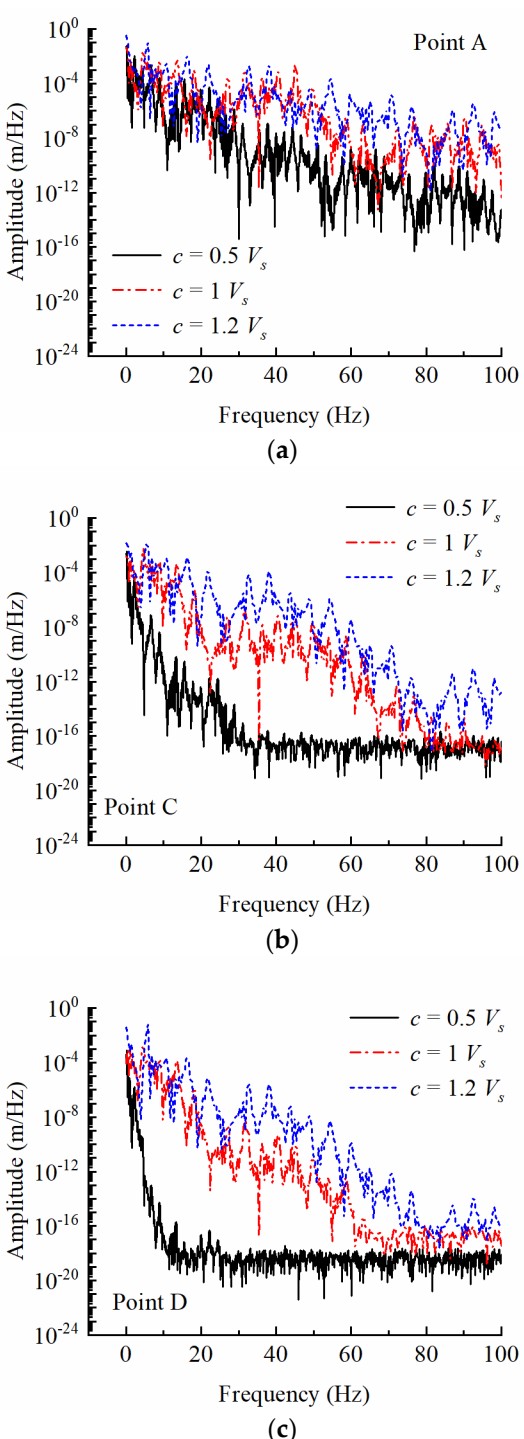

**Figure 10.** Frequency spectrum of vertical displacement for different train speeds at different observation points. (**a**) Point A; (**b**) Point C.; (**c**) Point D.

As shown in Figure 10, at point A, the frequency spectrum amplitudes for the three speeds are larger than other observation points. In the low-frequency region, the differences in the corresponding amplitudes under different velocities are not obvious. However, in the high-frequency region (>30 Hz), the amplitude increases with the velocity. Figure 10b,c illustrates that when the train runs at 0.5 $V_s$, the high-frequency response in the foundation attenuates rapidly, while the high-spectrum lines at the other two speeds remain in a higher position. This suggests that the high-frequency response at high speed attenuates slowly in

the foundation, leading to a larger high-frequency response area and a slower attenuation rate in the surrounding foundation as the train speed increases.

Figure 11 shows the frequency spectra at observation points A, C, and D at different groundwater levels and a train running speed of 100 m/s.

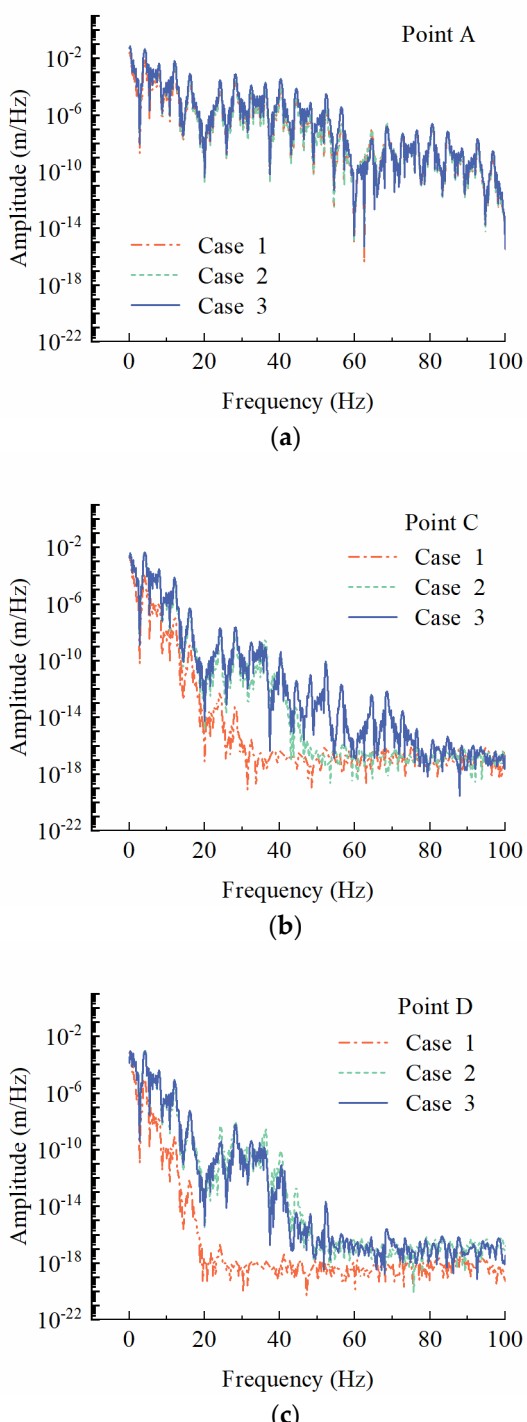

**Figure 11.** Frequency spectrum of vertical displacement for different groundwater levels of different points. (**a**) Point A; (**b**) Point C.; (**c**) Point D.

Figure 11a indicates that when the train runs at a speed of 100 m/s, there is almost no difference in amplitudes under different groundwater levels, suggesting that the rise of groundwater level has little impact on the spectral response of train load on the embankment surface. However, at observation points C and D, the frequency spectrum in

the high-frequency region increases significantly with the rise of the groundwater level, and the higher the groundwater level, the wider the frequency range. By comparing Figure 11a–c, it can be observed that the high-frequency response area caused by the train load on the surface of the embankment and foundation is affected by the groundwater level and distance from the route center. Specifically, the high-frequency response range increases with the rise of groundwater level, and it decreases as the distance from the center of the route increases.

## 5. Summary and Conclusions

Incorporating Biot's theory and employing the 2.5D finite element method, this study conducted an analysis of critical speed and vibration response of the track–embankment–foundation system under high-speed train conditions while also accounting for changes in groundwater level. The approach was validated by comparing results with model test results. The model has then been used to investigate the vibration responses to moving train loads, considering variations in groundwater levels and train speeds. The main findings are as follows:

(1) The critical velocity of the high-speed railway consistently decreases with the groundwater level rise. Moreover, the rise of the groundwater level within the embankment exerts a more pronounced influence on the system's critical velocity compared to the rise in groundwater level within the foundation. This underscores the significance of effective embankment waterproofing in controlling track vibrations;

(2) Train operations can induce deformation in both the embankment and foundation, with deformation significantly increasing as the groundwater level rises. In particular, when the groundwater level ascends from the foundation bottom to the subgrade surface, the deformation of the subgrade surface escalates by approximately 55%;

(3) The frequency spectrum of ground vibration increases significantly in the high-frequency region with the rising groundwater levels, and this increase affects a wider frequency range as the water level rises;

(4) This study indicates that the increase in groundwater level not only amplifies vibrations but also contributes to the extended propagation of high-frequency vibrations. Consequently, a more comprehensive analysis of the correlation between vibration propagation mechanisms and rising groundwater levels is imperative for future research;

(5) A limitation of this study is that the materials in the model are simulated using isotropic linear elastic properties. Future research could explore the anisotropic nature of materials and the polyphase composition of the media for a more comprehensive understanding.

**Author Contributions:** Conceptualization, J.H.; Methodology, J.H.; Validation, S.W.; Formal analysis, L.J.; Resources, B.Z.; Data curation, Y.T.; Supervision, J.H. and X.W.; Funding acquisition, J.H. All authors have read and agreed to the published version of the manuscript.

**Funding:** This research was funded by National Natural Science Foundation of China, grant number No. 52108308; Young Scientist Program of Fujian Province Natural Science Foundation, grant number No. 2020J05107; the Ministry of Education Key Laboratory of Soft Soils and Geoenviromental Engineering (Zhejiang University), grant number No. 2020P05. And The APC was funded by National Natural Science Foundation of China.

**Data Availability Statement:** The processed data cannot be shared at this time as the data also forms part of an ongoing study.

**Conflicts of Interest:** The authors declare no conflict of interest.

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
