# Peer review of "Effect of Groundwater Level Rise on the Critical Velocity of High-Speed Railway"

_water, doi:10.3390/w15213764_

Round 1

Reviewer 1 Report

The paper deals with the analysis, calculation, and, finally, determination of the impact of the groundwater on the critical velocity of high-speed railway construction. The idea is well-defined about the methodology, which is also finely stated. 

I am proposing a major revision.

After several readings, I have doubts about the numerical model's definition. Authors are defining unsaturated and saturated layers. What about the different levels of saturation? 

Are any field test results available, or is it planned to be done? The paper should incorporate such. 

Authors should enclose the map and description of the location where the analysis will be done. 

At least a picture of the train and railway construction view should be added. 

A literature review consists of mostly authors from China. I need clarification about the organized citation process. Authors should include authors from other countries because there are many similar models from other authors. 

Reviewer 2 Report

Review of manuscript titled "Effect of groundwater level rise on the critical velocity of high speed railway". The manuscript is well written and the modeling approach seems to be fine. The following are the comments that need to be addressed for possible acceptance of the manuscript.

1. The track geometry, such as the alignment of the track and the curvature of the track, can also affect the critical velocity. Why this is not considered in the study?

2. Biot's porous media theory is a linear theory. This means that it cannot be used to model the behavior of porous media under large deformations or high stresses. Also it is a homogeneous theory. This means that it cannot be used to model the behavior of porous media with non-uniform properties. Finally, this theory does not account for all of the physical processes that can occur in porous media. For example, it does not account for the effects of viscoelasticity, plasticity, or damage. How these limitations are addressed in your study?

3. Recheck the validity and constraints of equations 3,4,5 and 6. 

4. In Table 3, there is no information on Slab joints. Include it.

5. The conclusions section should be enhanced with further discussion and limitations of the study.

6. The abstract section should be rewritten so that it includes methodology and results.

Reviewer 3 Report

Please open the attached file

Round 2

Reviewer 1 Report

This time, the authors made a significant effort to improve the paper. The manuscript is ready for publication. 

Reviewer 2 Report

I congratulate the authors for revising the manuscript taking into consideration of all reviewers comments. I recommend the manuscript for publication.

Reviewer 3 Report

 The authors made all the required amendments, and therefore I recommend publishing the paper.

The English language is fine